# Glutathione Peroxidase GPX1 and Its Dichotomous Roles in Cancer

**DOI:** 10.3390/cancers14102560

**Published:** 2022-05-23

**Authors:** Yangjing Zhao, Hui Wang, Jingdong Zhou, Qixiang Shao

**Affiliations:** 1Jiangsu Key Laboratory of Medical Science and Laboratory Medicine, School of Medicine, Jiangsu University, Zhenjiang 212013, China; zhaoyangjing@ujs.edu.cn (Y.Z.); 1000004613@ujs.edu.cn (H.W.); 2Department of Hematology, Affiliated People’s Hospital of Jiangsu University, Zhenjiang 212002, China; 3Institute of Medical Genetics and Reproductive Immunity, School of Medical Science and Laboratory Medicine, Jiangsu College of Nursing, Huai’an 223005, China

**Keywords:** glutathione peroxidase 1, oxidative stress, gene polymorphism, cancer risks, prognostic biomarker, cancer

## Abstract

**Simple Summary:**

Glutathione peroxidase 1 (GPX1), as one of the most important antioxidant enzymes, maintains cellular redox balance by regulating reactive oxygen species levels. In this review, we summarize and discuss the relationship between GPX1 genetic polymorphisms and cancer susceptibility, as well as the dichotomous roles of GPX1 aberrant expressions in the occurrence and development of human malignancies via various biological mechanisms. Our work may deepen the understanding of the critical clinical values and future directions of GPX1 as a potential prognostic biomarker and novel cancer therapeutic target.

**Abstract:**

As the first identified selenoprotein, glutathione peroxidase 1 (GPX1) is a widely and abundantly expressed antioxidant enzyme. GPX1 utilizes glutathione as a substrate to catalyze hydrogen peroxide, lipid peroxide, and peroxynitrite, thereby reducing intracellular oxidative stress. The GPX1 gene is regulated at transcriptional, post-transcriptional, and translational levels. Numerous case-control studies and meta-analyses have assessed the association between a functional genetic polymorphism of the GPX1 gene, named Pro198Leu (rs1050450 C>T), and cancer susceptibility in different populations. GPX1 polymorphism has type-specific effects as a candidate marker for cancer risk, but the association between GPX1 variants and cancer susceptibility remains controversial in different studies. GPX1 is abnormally elevated in most types of cancer but has complex dichotomous roles as tumor suppressor and promoter in different cancers. GPX1 can participate in various signaling pathways to regulate tumor biological behaviors, including cell proliferation, apoptosis, invasion, immune response, and chemoresistance. In this review, we comprehensively summarize the controversial associations between GPX1 polymorphism and cancer risks and further discuss the relationships between the aberrant expressions of GPX1 and tumorigenesis. Further studies are needed to elucidate the clinical significance of GPX1 as a potential prognostic biomarker and novel therapeutic target in various malignancies.

## 1. Introduction

Proper redox balance is strictly regulated and critical for organismal homeostasis maintenance and cellular physiological functions. Reactive oxygen species (ROS), including superoxide (O_2_^.^^−^) and hydrogen peroxide (H_2_O_2_), are an array of derivatives of molecular oxygen with higher reactivity than molecular oxygen (O_2_) and are produced from aerobic metabolism [1]. ROS plays an important role in maintaining cellular signal transduction and homeostasis at physiological levels. However, if ROS levels increase dramatically and exceed the capacity of cellular antioxidant defense systems, ROS can induce significant damage to cellular structure and function, termed oxidative stress [2]. The antioxidant defense systems contain three layers: small molecular antioxidants, antioxidant enzymes, and damage-removing or repairing enzymes. As the intermediate defense against ROS, the major antioxidant enzymes include superoxide dismutases (SOD), catalase (CAT), glutathione peroxidases (GPXs), and peroxiredoxins (Prxs) [3]. GPXs are an antioxidant enzyme family with peroxidase activity to catalyze the reduction of H_2_O_2_ and lipid hydroperoxides by converting glutathione (GSH) to oxidized glutathione (GSSG), thereby protecting the organism against oxidative damage [4]. Human GPXs are composed of eight isozymes (GPX1-8), of which five members (GPX1-4 and 6) are selenocysteine (Sec)-containing proteins (SecGPX), and three members (GPX5, 7, and 8) are cysteine-containing proteins (CysGPX) with the active site Sec replaced by cysteine. These isozymes are encoded by different genes and differ in molecular structure, subcellular localization, substrate specificity, enzyme attributes, and biological function [5]. For example, GPX4 is a widely studied phospholipid, hydroperoxidase, with a high preference for lipid hydroperoxides to protect cells from membrane lipid peroxidation and ferroptosis [4]. GPX family members have been reported to be involved in a variety of diseases, including neurodegenerative disorders, diabetes, cardiovascular disease, and cancer [6].

As the first discovered selenoprotein in 1973, GPX1 is the most abundant GPX isoform. It is ubiquitously expressed in many tissues and mainly distributed in the cytosol and mitochondria of the erythrocytes, liver, lungs, and kidney [7,8]. During the past few decades, many researchers have explored the gene expressions and biochemical functions of GPX1 and revealed its roles in the occurrence and process of various diseases, such as metabolic diseases [9], neuropsychiatric disorders [10], lung inflammation [11], and tumors [12]. Many studies have already reported that GPX1 gene polymorphisms have complex associations with cancer risks and patient survival. Moreover, aberrant expression of GPX1 in multiple cancers is closely related to oncogenesis and cancer progression [13,14]. This review aims to summarize the molecular structure, function of GPX1, and its roles in redox regulation and further highlight the studies focused on its roles and underlying mechanisms in multiple human cancers to evaluate its therapeutic and prognostic values for cancers.

## 2. The Molecular Characteristic and Function of GPX1

### 2.1. GPX1 Molecular Structure

GPX1 gene is located on chromosome 3p21.31 with 1178 base pairs in size and contains two exons. It has five transcript variants, and its transcription direction is negative (Figure 1A,B) [4]. The purified active mammalian GPX1 protein is a homotetramer consisting of four identical subunits with a molecular weight of 22–23 kDa. Each monomer has about 208 amino acids (Figure 1C) [15]. As selenoprotein, the selenium present in GPX1 protein is generated by the insertion of the 21st amino acid selenocysteine into the nascent polypeptide chain during the process of translational coding at the UGA stop codon. The coding sequence of the selenocysteine codon is in exon 1. Generally, the UGA stop codon sequence terminates translation in non-selenocysteine-encoding proteins. In contrast, the 3’ UTR of selenoprotein mRNA contains a conserved stem-loop structured cis-acting element called Sec insertion sequence (SECIS), or Sec insertion sequence, which is required for the recognition of UGA as a Sec codon rather than a stop signal [16,17]. Like other glutathione peroxidases, GPX1 has a conserved catalytic tetrad composed of Sec or Cys, Glutamine (Gln), Tryptophan (Trp), and Asparagine (Asn). The reactive Sec in GPX1 protein is surrounded by four arginines and one lysine of an adjacent subunit, and these five residues are thought to bind glutathione [5].

### 2.2. Regulation of GPX1 Expression and Activity

GPX1 can be regulated by transcriptional, post-transcriptional, and translational levels. Several investigators have shown that GPX1 expression is mainly regulated by several transcription factors and oxygen tension at the transcriptional level. The ETS-domain transcription factor PU.1 can directly bind to the promoter (−800 region) and the 3’ flanking sequence (+1010 region) of GPX1. PU.1 transcriptionally regulates GPX1 mRNA expression in various myeloid and lymphoblastoid lineages, which are vital for neutrophil and macrophage maturation [18]. Under normal cell culture conditions, tumor suppressor p53 can bind to the GPX1 promoter (−694~−720 region) and activate GPX1 expression and its antioxidant response. P53 can simultaneously activate the expressions of H_2_O_2_-producing enzyme manganese superoxide dismutase (MnSOD) and antioxidant enzyme GPX1, but not catalase. These imbalanced antioxidant enzymes induce oxidative stress and apoptosis [19,20]. High glucose can induce TBP-associated factor 1 (TAF1)-mediated p53 Thr55 phosphorylation in endothelial cells, which dissociates p53 from the GPX1 promoter and leads to GPX1 expression decreases and oxidative stress [21]. In skeletal muscle cells, the transcription factors nuclear factor kappa B (NF-kB) and activating protein 1 (AP-1) regulate oxidant-induced GPX1 expression upregulation. NF-kB transcription complexes include a p50/p65 heterodimer, a p50 homodimer, and a p50/RelB heterodimer, while AP-1 complexes contain a c-jun/c-fos heterodimer [22]. There is an important antioxidant survival pathway in mitochondrial-defective cells that the up-regulated transcription factor zinc finger protein 143 (ZNF143) transcriptionally activates GPX1 activity and thereby protects cells from oxidative damage [23]. In addition, there are two oxygen-responsive elements (ORE) in the GPX1 promoter region. Oxygen-responsive element-binding protein (OREBP) can bind to them to regulate GPX1 transcription in response to oxygen tension [24,25].

GPX1 is post-transcriptionally regulated by the presence or absence of selenium and cofactors involved in Sec biosynthesis and insertion processes. Selenocysteine insertion into GPX1 proteins involves a special encoding mechanism for recognizing the UGA stop codon. The UGA codon serving as a selenocysteine insertion signal in selenoproteins is mainly mediated by the SECIS element [26]. SECIS-binding protein 2 (SBP2) binds to the SECIS element and recruits the selenate-cysteine-specific elongation factor (EFsec) complex along with Sec-specific tRNA (tRNA^Sec^) to the ribosome, with L30 serving as an anchor. Thus, the SECIS element and SBP2 serve as the core components of UGA recoding factors and determine the selenocysteine insertion efficiency for GPX1 [27]. GPX1 translation is altered upon experimental selenium depletion and alteration of the SECIS machinery at the translational level. GPX1 is mainly regulated by the sequence and structure of SECIS element, SBP2, tRNA^Sec^, and elongation factors to regulate Sec incorporation. Sec incorporation and GPX1 protein synthesis are inhibited by homocysteine because that Cys incorporation is more efficient than Sec in the translational process (Figure 2). Elevated levels of homocysteine in vivo and in vitro models are observed to downregulate GPX1 translation and activity, but not GPX1 transcription, by interfering with the essential synthesis mechanism for selenocysteine-containing protein [28].

At the GPX1 locus, there is a selenocysteine codon sequence in exon 1, a Sec insertion sequence (SECIS) in the 3’ UTR, and two oxygen-responsive elements (ORE) in the promoter. The yellow circles represent the transcription factors or DNA-binding proteins that bind to the GPX1 locus, which participate in the transcriptional, post-transcriptional, and translational regulation of GPX1. NF-kB: nuclear factor kappa B, AP-1: activating protein 1, ZNF143: zinc finger protein 143, Nrf1/2: Nuclear respiratory factor, OREBP: Oxygen-responsive element-binding protein, SBP2: SECIS-binding protein 2.

### 2.3. Enzymatic Mechanisms of GPX1

GPX1 is one of the most abundantly expressed members of GPXs and is ubiquitously expressed in all cells, mainly located in the cytoplasm, mitochondria, nucleus, and peroxisomes [5]. The superoxide radicals (O_2_^.^^−^) are often generated from normal oxygen metabolism, and can also be catalyzed by various enzymes, such as NADPH oxidase (NOX) and uncoupled endothelial nitric oxide synthase (eNOS). O_2_^.^^−^ mainly exists in the extracellular matrix, cytoplasm, and some organelles (e.g., mitochondria and peroxisome). O_2_^.^^−^ can be metabolized to water by a two-step catalytic reaction. O_2_^.^^−^ is first converted into hydrogen peroxide and then into the water via the catalysis of various enzymes. The catalytic enzymes for O_2_^.^^−^ in different subcellular localizations are different. O_2_^.^^−^ is catalyzed to hydrogen peroxide by superoxide dismutase (ECSOD) in the extracellular matrix, by copper or zinc superoxide dismutase (CuSOD, ZnSOD) in the cytoplasm, and by MnSOD in the mitochondria. Hydrogen peroxide is then enzymatically reduced by GPX1, Prx, and catalase. It is mainly catalyzed by GPX1 and Prx 1,2 in the cytoplasm, GPX1 and Prx 3,5 in the mitochondria, and catalase and GPX1 in the peroxisome. Under certain oxidative stress conditions, hydrogen peroxide can undergo a Fenton reaction with free iron (Fe^2+^) to generate harmful hydroxyl radicals (OH) and the downstream lipid hydroperoxides (LOOH) and induce lipid peroxidative damage. In addition, O_2_^.^^−^ can interact with nitric oxide (NO) to form peroxynitrite radicals (ONOO^−^). The antioxidant effects of GPX1 are achieved through the direct reduction of hydrogen peroxide, lipid hydroperoxide, and oxynitride radicals. In each GPX1 catalyzed reaction, two reduced GSH are consumed to generate one GSSG (Figure 3) [13,29]. GPX1 also catalyzes a three-step circular reaction of the decomposition process of peroxides (Figure 3). The decomposition reaction involves the formation of some stable intermediary modifications at the active-site selenocysteine. Active GPX1-SeH reacts with peroxide (ROOH or hydrogen peroxide) to generate selenic acid (SeOH). One molecule of GSH reduces selenic acid to GPX1-Se-SG and water. Another molecule of GSH then converts the GPX1-Se-SG enzyme into GPX1-SeH and GSSG (Figure 3) [13]. 

As the main form of ROS, O_2_^.^^−^ is generated from normal cellular metabolic processes or enzyme sources, such as NADPH oxidase (NOX) and endothelial nitric oxide synthase (eNOS). All O_2_^.^^−^ localized to the extracellular, cytosol, mitochondria, or peroxisome compartments, is neutralized to water through a two-step enzymatic reaction. The first step is that various superoxide dismutases catalyze O_2_^.^^−^ to produce hydrogen peroxide. Superoxide dismutases include superoxide dismutase (ECSOD), copper or zinc superoxide dismutase (CuSOD, ZnSOD), and manganese superoxide dismutase (MnSOD). The second step is mediated by GPX1, peroxidase (Prx), and catalase to catalyze the decomposition of hydrogen peroxide into water. GPX1 typically uses glutathione (GSH) as a reducing agent to reduce hydrogen peroxide to generate oxidized glutathione (GSSG) and water. Thus, GPX1 plays a fundamental role in metabolizing intracellular ROS by reducing hydrogen peroxide, lipid hydroperoxides (LOOH), and peroxynitrite radicals (ONOO^−^). The three-step cyclic reaction shows decomposition process of peroxides under GPX1 catalysis and its stable intermediary modifications at the active-site selenocysteine. The net reaction is summarized in the red box.

## 3. GPX1 and Cancer

### 3.1. GPX1 Gene Polymorphisms and Cancer Susceptibility

Human gene polymorphism is a common phenomenon, but the distribution frequency of genetic polymorphism varies among races. Genetic polymorphism is defined as a heterozygous DNA sequence variation with a frequency greater than 1% in the general population. Genetic polymorphisms can be classified as single nucleotide polymorphism (SNP) with a single nucleotide variation and length polymorphism with many repetitive DNA sequence changes, such as minisatellites or microsatellites. Most polymorphisms are silent, but some polymorphisms can lead to aberrant protein expressions and structures. Thus, functional gene polymorphisms can become the molecular basis of disease pathogenesis as disease-inducing factors closely related to the susceptibility of various diseases, especially tumors [30].

The human GPX1 gene has several genetic polymorphisms. The most common functional polymorphism of GPX1 is a cytosine (C) to thymine (T) substitution in the exon 2 at codon 198 (Pro198Leu, dbSNP ID: rs1050450), resulting in amino acid variations from proline (Pro, CCC) to leucine (Leu, CTC) and GPX1 enzymatic activity declines by 5%. The population can be divided into three genotypes according to GPX1 Pro198Leu polymorphism: Pro/Pro homozygotes, Pro/Leu heterozygotes, and Leu/Leu homozygotes [31]. As proline is the only amino acid that does not have a free unsubstituted amino group on the alpha carbon atom, Pro to Leu substitution is speculated to lead to a conformational change of GPX1 protein [32,33]. During the past two decades, GPX1 Pro198Leu polymorphism has been extensively studied in various cancer types [34], mainly breast [35], bladder [36], prostate [37], lung [38], leukemia [39], and colon cancers [40]. However, the associations between GPX1 genetic variants and cancer susceptibility are still controversial and inconclusive due to the differences in study cohorts and statistical methods (Table 1). A comprehensive meta-analysis including 31 published articles found that GPX1 Pro198Leo polymorphism may promote cancer susceptibility by disturbing the antioxidant balance. Variant Leu-allele carriers (Pro/Leu and Leu/Leu) have increased cancer risk, especially in Asian subgroups in a dominant genetic model [41]. However, another meta-analysis including 35 published articles suggested no associations between GPX1 Pro198Leu polymorphism and cancer risk in articles with high-quality criteria. In contrast, strong associations were identified in articles with low-quality criteria [42].

#### 3.1.1. GPX1 Polymorphism and Breast Cancer Susceptibility

In 2004, Cox D et al. observed no associations between the Pro198Leu polymorphism (Leu198 allele) or −1040 G/A polymorphism (1040A allele) of GPX1 and breast cancer risks in the Caucasian female population [43]. In 2018, Habyarimana T et al. examined the presence and genotype distributions of GPX1 Pro198Leu polymorphism in the Rwandan population. They found that it was not a risk factor for breast cancer in Rwandans [44]. However, in 2003, Hu Y et al. analyzed the frequency of GPX1 Pro198Leu polymorphism. It was discovered that the leucine-containing allele was more frequently associated with breast cancer than the proline-containing allele and with lower GPX1 enzyme activity to selenium-mediated stimulation [45]. A similar trend is also observed in the Danish female population. In 2006, studies by Ravn H showed that carriers with the variant T allele of GPX1 Pro198Leu had a higher risk of breast cancer and decreased GPX1 activity than non-carriers in Danish women using prospectively collected blood samples [46]. In contrast, in 2010, Ermolenko N et al. reported that the variant T allele of GPX1 Pro198Leu was protective against sporadic breast cancer in residents of the Altai Krai region of Russia [47]. In 2015, Jablonska E et al. conducted genotype stratification in Polish women and found that breast cancer patients with the GPX1 Pro/Pro homozygote had the highest GPX1 activity among all genotypes. Patients with the GPX1 Pro/Pro homozygote showed a significant correlation between GPX1 activity and lipid peroxidation level, not observed in any other genotypes. They proposed that the GPX1 variant Leu-allele has a protective effect against breast cancer, and Pro198Leu polymorphism is significantly associated with breast cancer risk via affecting oxidative stress [31]. Considering the controversial effects of GPX1 polymorphism on breast cancer, a meta-analysis including six case-control studies was performed and revealed no significant correlation between GPX1 Pro198Leu polymorphism and breast cancer susceptibility in overall populations. However, subgroup analysis showed that variant Leu-allele elevates cancer risk in the African population, which still needs larger-scale surveys to confirm [35]. Several researchers expressed GPX1 protein encoded by allelic variants in human lymphoblast and breast cancer cell lines and found that alleles expressing Pro198Leu and Ala7 polymorphisms were more cytoplasmically located than other alleles. Artificially altering the expression and distribution of GPX1 in the cellular cytoplasm and mitochondria leads to changes in oxidative stress, energy metabolism, and cancer signaling pathways, suggesting that genetic polymorphism and cellular location of GPX1 can affect cell biology [48,49].

#### 3.1.2. GPX1 Polymorphism and Bladder Cancer Susceptibility

As with breast cancer, there is disagreement about whether GPX1 polymorphism is associated with bladder cancer susceptibility. It was reported that GPX1 Pro198Leu polymorphism is not associated with the occurrence, progression, or stage of bladder cancer in Egyptian and Moroccan populations [50,51]. In 2018, Nikic P et al. reported that GPX1 polymorphism was not associated with the overall survival (OS) of patients with metastatic urothelial bladder cancer treated with cisplatin-based chemotherapy in the Serbian population [52]. However, individuals with GPX1 Pro198Leu polymorphism have a significantly higher probability of developing bladder cancer than controls in Japanese, Ecuadorian, and Turkish populations [53,54,55]. Some systematic reviews and meta-analyses demonstrated that GPX1 Pro198Leu polymorphism significantly increased bladder cancer susceptibility [36,56]. In 2017, Wang C et al. conducted a comprehensive meta-analysis including 60 case-control studies and 52 published articles to reveal that GPX1 Pro198Leu polymorphism was only positively associated with the development and progression of bladder, head/neck, and brain cancers among all tumor types [34]. In contrast, in 2005, Zhao H et al. reported that the GPX1 variant genotype had a borderline protective role in superficial bladder cancer to extend overall recurrence-free survival times. Thus, Pro198Leu polymorphism may serve as a molecular marker for monitoring bladder cancer recurrence [57].

#### 3.1.3. GPX1 Polymorphism and Prostate Cancer Susceptibility

In 1994, Moscow J et al. described a trinucleotide polymorphism (GCG repeats, alanine codon repeats) in GPX1 exon 1, encoding five to seven alanine residues for the polyalanine track (Ala5, Ala6, Ala7) [58]. In 2002, Kote-Jarai Z et al. tested the frequency of the GCG repeat polymorphism of GPX1 in Caucasians. They found that the Ala6/Ala6 genotype had increased frequency in prostate cancer but had no association with prostate cancer risk [59]. Two studies have demonstrated that there is no difference in the genotype frequency of GPX1 Pro198Leu polymorphism between prostate cancer patients and healthy controls or between aggressive and non-aggressive prostate cancer patients in the Turkish population [60,61]. Several meta-analyses also found that GPX1 gene polymorphism was not significantly associated with prostate cancer susceptibility in dominant, recessive, and codominant genetic models [36,62]. In contrast, in 2009, Arsova-Sarafinovska Z et al. found that the variant Leu-allele of GPX1 polymorphism had an overall protective effect on prostate cancer risk. The risk of prostate cancer in heterozygous carriers with variant Leu alleles is lower than that in homozygous wild-type individuals [37].

#### 3.1.4. GPX1 Polymorphism and Other Types of Cancer Susceptibility

In 2000, Ratnasinghe D et al. found that the heterozygous variant of GPX Pro198Leu polymorphism significantly increased the risk of lung cancer in Caucasians but not in Chinese [33]. However, Raaschou-Nielsen O et al. found that participants with more GPX1 variant alleles (GPX1^TT^ variant genotype) had a statistically significant lower risk of lung cancer [38]. In 2005, Jefferies S et al. proposed that GPX1 Ala6 and Ala7 alleles were significantly associated with squamous cell carcinoma development [63]. No associations were observed in any of the investigated cases between GPX1 Pro198Leu polymorphism and colorectal cancer risk in both Norwegian [40] and Danish [64] populations, endometrial cancer [65], urothelial cancer [66], or chronic myeloid leukemia [39]. Additionally, there was no significant association between GPX1 Pro198Leu polymorphism and papillary thyroid carcinoma or its demographic, clinical, and pathological characteristics [67].

**Table 1 cancers-14-02560-t001:** The associations between GPX1 polymorphisms and cancer risks.

Tumor Type	Study Cohort (Race/Country)	GPX1 Polymorphism	Sample Type	Associations with Cancer risk	Refs.
Breast cancer	Caucasian, Rwanda	Pro198Leu -1040 G/A, Pro198Leu	Unknown, Peripheral blood	No association	[43,44]
	Chicago/America, Denmark	Pro198Leu	Peripheral blood	Variant Leu-allele is associated with higher cancer risk and lower enzyme activity	[45,46]
	Meta-analysis	Pro198Leu	—	No association in Caucasians; Variant Leu-allele increases breast cancer risk in African population	[35]
	Altai krai/Russia, Poland	Pro198Leu	Unknown, Peripheral blood	Variant Leu-allele decreases sporadic breast cancer risk	[31,47]
Bladder cancer	Egypt, Morocco, Serbia	Pro198Leu	Unknown, Peripheral blood	No association	[50,51,52]
	Japan, Turkey, Ecuador	Pro198Leu	Peripheral blood, Tumor tissues in paraffin	Variant Leu-allele increases bladder cancer risk	[53,54,55]
	Meta-analysis	Pro198Leu	—	Variant Leu-allele increases bladder cancer risk	[34,36,56]
	America	Pro198Leu	Peripheral blood	Variant Leu-allele decreases bladder cancer risk	[57]
Prostate cancer	Turkey	Pro198Leu	Peripheral blood	No association	[60,61]
	Meta-analysis	Pro198Leu	—	No association	[36,62]
	Caucasian/England	GCG repeat	Peripheral blood	Ala6/Ala6 genotype increases in prostate cancer, but has no association with cancer risk	[59]
	Macedonia	Pro198Leu	Peripheral blood	Variant Leu-allele decreases prostate cancer risk	[37]
Lung cancer	Caucasians and Chinese	Pro198Leu	Unknown	Variant Leu-allele increases bladder cancer risk in Caucasians but not Chinese	[33]
	Denmark	Pro198Leu	Frozen lymphocytes	Variant Leu-allele decreases lung cancer risk	[38]
Head and neck cancer	Meta-analysis	Pro198Leu	—	Variant Leu-allele increases head and neck cancer risk	[34]
	England	GCG repeat	Peripheral blood	Ala6 and Ala7 genotypes increase in head and neck cancer risk	[63]
Brain cancer	Meta-analysis	Pro198Leu	—	Variant Leu-allele increases bladder cancer risk	[34]
Colorectal cancer	Norway, Denmark	Pro198Leu	Peripheral blood	No association	[40,64]
Endometrial cancer	Poland	Pro198Leu	Peripheral blood	No association	[65]
Urothelial tumors	Balkan	Pro198Leu	Peripheral blood	No association	[66]
Chronic myeloid leukemia	Romania	Pro198Leu	Peripheral blood	No association	[39]
Papillary thyroid carcinoma	America	Pro198Leu	Peripheral blood	No association	[67]

### 3.2. GPX1 Altered Expression and Activity in Cancer

It is now recognized that GPX1 is closely related to tumorigenesis and development, mainly due to its role in eliminating hydroperoxides by regulating intracellular reactive oxygen species. A recent study conducted by Wei R used bioinformatics methods to comprehensively analyze GPX1 expression levels and prognostic values in different human malignancies. Transcriptome-sequencing data from the Cancer Genome Atlas (TCGA) public database revealed that GPX1 is highly expressed in most cancers and has higher expressions in tumor tissues than adjacent normal controls. These include glioblastoma multiforme, renal papillary cell carcinoma (KIRP), acute myeloid leukemia (AML), low-grade glioma (LGG), ovarian serous cystadenocarcinoma, pancreatic adenocarcinoma, skin melanoma, testicular germ cell tumor, thyroid cancer, and endometrial cancer [68]. However, GPX1 may play opposite roles in different types of cancers, and its roles vary in different literature (Table 2).

#### 3.2.1. GPX1 as Tumor Suppressor

Pancreatic cancer

There appears to be a gradual decrease in antioxidant enzyme expression in pancreatic cells from a normal pancreas to chronic pancreatitis to pancreatic cancer [69]. GPX1 immunoreactivity was also decreased in four pancreatic cancer cell lines when compared to a normal pancreas. GPX1 levels in both human pancreatic cancer specimens and cell lines are also lower than those in a normal pancreas [70]. GPX1 overexpression slows pancreatic tumor growth in vitro and in vivo and GPX1 gene delivery is proven to be beneficial for pancreatic cancer treatment [71]. Tissue microarray analysis showed that GPX1 expression is significantly down-regulated in pancreatic cancer tissues compared with adjacent tissues, predicting poor patient prognosis. Moreover, GPX1 is downregulated in most pancreatic cancer cell lines compared with pancreatic ductal epithelial cells. Both in vitro and in vivo assays demonstrated that GPX1 silencing drives a mesenchymal transition phenotype and gemcitabine resistance by activating the ROS-mediated Akt/GSK3β/Snail signaling axis in pancreatic cancer cells [72]. Pancreatic ductal adenocarcinoma cells can induce protective autophagy via the activation of ROS/AMPK signaling and GPX1 degradation to survive in a glucose-starved tumor microenvironment. Both GPX1 overexpression and autophagy inhibition can sensitize cells to starvation-induced cell death via the activation of caspase-dependent apoptosis [73]. These studies suggest that GPX1 is down-regulated in pancreatic cancer cells and clinical tissues and may act as a tumor suppressor gene in pancreatic cancer.

Other cancers

Loss (reduction) of GPX1 mRNA and protein expressions due to aberrant hypermethylation of the promoter region in gastric cancer cell lines and tissues is significantly associated with aggressiveness and poor survival in gastric cancer patients [76]. In 2018, Metere A et al. found that compared to healthy cells, GPX1 protein expression decreases, and free radicals increase in thyroid tumor tissues by detecting tissues from total thyroidectomy, indicating an unbalanced oxidation/antioxidant system in thyroid cancer tissues [77].

#### 3.2.2. GPX1 as Tumor Promoter

Breast cancer

GPX1 overexpression in breast cancer cell line T47D can partially inhibit doxorubicin-induced apoptosis by interfering with the sphingomyelin-ceramide pathway [80]. Docosahexaenoic acid dietary supplementation increases tumor sensitivity to anthracyclines by reducing cytosolic GPX1 activity in breast cancer cell line MDA-MB-231 and rat mammary tumors, which can be abolished by vitamin E [81]. GPX1 expression is absent in non-triple negative breast cancer (TNBC) cells but abundant in TNBC cells. GPX1 depletion prevents cell migration and invasion by downregulating FAK/c-Src activation in vitro and reduces lung metastasis in vivo [82]. GPX1 silencing in RIPK3-negative cancer cells significantly increases H_2_O_2_ levels, sustains the activation of JNK and Caspase-8, and promotes TNF-α-induced apoptosis. GPX1 knockdown promotes apoptosis and reduces the tumorigenic growth of breast cancer cell line MDA-MB-231 [83]. The mitochondrial enzyme glutamate dehydrogenase 1 is upregulated in various human cancers, maintaining the level of its subsequent metabolite fumarate. Fumarate can activate GPX1 to perturb redox homeostasis and promote cell proliferation of breast and lung cancer cells [84]. Therefore, GPX1 may exhibit tumor-promoting activity in breast cancer mainly by regulating cancer cell apoptosis and redox.

Other cancers

In freshly isolated hepatocellular carcinoma tissues, decreased expression of selenium-binding protein 1 promotes tumor vascular invasion by increasing GPX1 activity [86]. In colorectal cancer cell DLD-1, transforming growth factor-β1 (TGF-β1) protects cancer cells from exogenous H_2_O_2_-induced oxidative damage and cell death by upregulating GPX1 protein expression and enzymatic activity [87]. In esophageal cancer and salivary adenoid cystic carcinoma cells, GPX1 expression can promote invasion, migration, proliferation, and cisplatin resistance. Since NF-κB can transcriptionally activate GPX1, vitamin D can inhibit the NF-κB pathway and GPX1 expression to reduce tumor malignancy [88,89]. Downregulated miR-153 expression in glioma stem cells (GSCs) promotes its target gene, nuclear factor-erythroid 2-related factor-2 (Nrf-2), which cascades to activate GPX1 transcription and reduce ROS levels, resulting in further enhancement of the radioresistance and stemness in GSCs. Therefore, GPX1 mRNA, protein, and enzyme activities in GSCs are all significantly higher than those in non-GSCs glioma cells [85]. GPX1 expression is upregulated in non-small cell lung cancer (NSCLC) cell lines resistant to cisplatin. GPX1 overexpression significantly inhibits cisplatin-induced intracellular ROS accumulation, activates the PI3K-AKT pathway, and further leads to cisplatin resistance in NSCLC cells [90].

### 3.3. Prognostic Values of GPX1

GPX1 is closely related to tumor development, patient survival, and prognosis in a variety of human malignancies [68]. However, the expression levels and prognostic values of GPX1 in different types of tumors are still controversial according to different cohorts and statistical analyses [78,79,92]. Bioinformatics analysis of RNA sequencing data from 31 types of cancers in TCGA revealed that high GPX1 expression levels indicate worse OS and disease-free survival (DFS) prognosis in LGG and AML but better OS in KIRP [68]. Functional enrichment analysis found that GPX1 is involved in several important biological processes, including ROS metabolic process, ferroptosis, cell growth signal transduction, glutathione metabolism, and p53-regulated metabolic pathways [68]. Similarly, Zhang J and coworkers analyzed the RNA sequencing data of 151 AML patients in TCGA combined with complete clinical information. They verified the results with the Gene Expression Omnibus (GEO) dataset and in vitro experiments. They comprehensively analyzed the expression, prognostic value, and potential pathogenic processes of GPX1 in AML. They found that high expression of GPX1 in AML patients was associated with unfavorable prognosis and constructed a GPX1-associated prognostic signature as an independent unfavorable prognostic factor for AML. Moreover, GPX1 may be involved in AML immunosuppression for its positive correlations with immunosuppressive cells (e.g., myeloid-derived suppressor cells and regulatory T cells), immune inhibitory checkpoint-mediated pathways (TIM3/Galectin-9 and SIRPα/CD47 pathways), and negative correlations with low fraction levels of CD4^+^ and CD8^+^ effector T cells [92]. 

High GPX1 level in patients with renal cell carcinoma (RCC) is positively associated with poor OS, distant metastasis, lymph node metastasis, and tumor stage [79]. Another study demonstrated that GPX1 is upregulated in three types of RCC, including KIRP, chromophobe carcinoma (KICH), and renal clear cell carcinoma (KIRC). However, high GPX1 expression suggests an adverse prognosis in KICH and KIRC patients but a favorable prognosis in KIRP patients [78]. Immunohistochemical analysis based on large clinical samples found that high GPX1 expression in oral squamous cell carcinoma is significantly associated with lymph node metastasis, advanced stage, high grade, and invasion and predicts unfavorable patient survival [75]. However, immunohistochemical experiments conducted by Fu T et al. draw the opposite conclusion that the high expression level of GPX1 suggests better disease-specific survival in patients with buccal mucosal squamous cell carcinoma [74]. High expression of GPX1 in laryngeal squamous cell carcinoma is significantly correlated with lymph node metastasis and TNM stage (Tumour-Node-Metastasis), which can be applied as a negative prognostic factor for patient survival [91]. Microarray assay of prostate cancer tissues revealed that the quantification of GPX1 levels in different subcellular distributions shows no association with cancer recurrence [95]. GPX1 mRNA level is significantly elevated in malignant pleural mesothelioma tissues compared to normal adjacent pleural tissues, and the high GPX1 level is associated with a short survival time [94]. Current studies collectively indicate that high expression levels of GPX1 may have different prognostic values for different types of cancers.

### 3.4. GPX1 Inhibitors and Therapeutic Potential

The overexpression of GPX1 in various tumors can eliminate ROS, weaken apoptosis, and induce drug resistance, promoting cancer cell survival [80,83,90]. Hence, many studies are devoted to exploring GPX1 inhibitors and their applications for cancer therapy. There is no specific inhibitor for GPX1 because the GPX1 active site is structurally similar to other GPX members. Mercaptosuccinic acid (MSA), the most potent and best-characterized inhibitor of GPX1, can compete with glutathione for binding to the Sec active site of GPX1 [96]. A novel GPX1 inhibitor, Pentathiepins, has much stronger enzyme inhibitory activity to GPX1 than MSA. Pentathiepins can effectively inhibit GPX1 enzymatic activity and induce a loss of mitochondrial membrane potential and oxidative stress in cancer cells, resulting in DNA strand breaks and apoptosis. Pentathiepins show potent anticancer activity in various human cancer cell lines [97]. The combined treatment of GPX1 inhibitors, including MSA or Pentathiepins, with photodynamic therapy can generate synergistic anticancer effects by enhancing oxidative stress, accumulating ROS, and inducing apoptosis in tumor cells [98]. The increased GPX1 activity in tumor cells may contribute to resistance to antitumor drug therapy. GPX1 expression is significantly increased in the resistant cells isolated from lymphoma patients with clinical chemoresistance to methotrexate and etoposide. An acylhydrazone heterocycle with GPX1 inhibitory activity has been identified by library screen and crystal structure analysis, which can be used in combination with anticancer drugs to reverse therapy resistance in lymphoma cell lines [93].

## 4. Discussion and Perspectives

GPX1 is a major antioxidant enzyme that protects cells against lethal oxidative stress and maintains redox balance by regulating intracellular ROS levels. It is now well recognized that GPX1 is closely linked to the pathogenicity of various types of cancer. Although great efforts have been made to explore the association between GPX1 Pro198Leu gene polymorphism and cancer risks, the role of GPX1 polymorphism is yet to be further elucidated. There is still no consensus on whether GPX1 Pro198Leu gene polymorphism has deleterious effects, protective effects, or no association with cancer risks (Table 1). Currently, several studies have supported the role of GPX1 as a tumor suppressor in pancreatic cancer. However, GPX1 is abnormally overexpressed in most types of cancer and acts as a tumor promoter by regulating the proliferation, invasion, migration, apoptosis, immune response, and drug sensitivity of tumor cells. GPX1 has been proven to be a promising prognostic biomarker and is closely associated with clinicopathological features. It is currently known that the GPX1 expression level in tumor tissues is negatively correlated with the overall survival time of patients with breast, gastric, and glioma cancers and leukemia but is correlated with a favorable prognosis in pancreatic cancer patients (Table 2). Future studies should further explicit and develop the clinical values of GPX1 as a promising diagnostic biomarker. In addition, understanding the molecular mechanisms and functional networks responsible for the role of GPX1 in carcinogenesis is critical for developing novel strategies for precision cancer therapy.

## Figures and Tables

**Figure 1 cancers-14-02560-f001:**
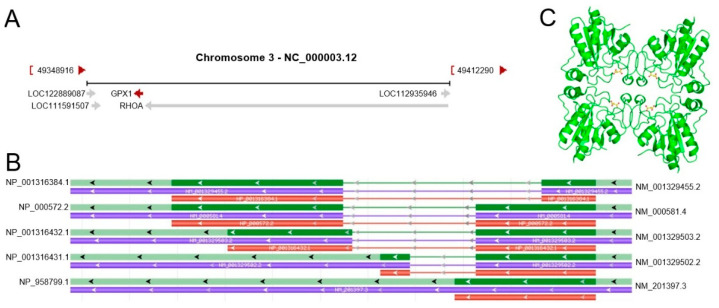
Transcription direction and gene structure of GPX1. (**A**) The red arrow indicates the negative transcription direction of GPX1. (**B**) The five transcript variants and their corresponding isoforms of GPX1 present in Genome Data Viewer (https://www.ncbi.nlm.nih.gov/genome/gdv/browser/gene/?id=2876, accessed on 1 May 2022). Isoform 1: NM_000581.4→NP_000572.2; isoform 2: NM_201397.3→NP_958799.1, isoform 3: NM_001329502.2→NP_001316431.1; isoform 4: NM_001329503.2→ NP_001316432.1; isoform 5: NM_001329455.2→NP_001316384.1. (**C**) The front view of the crystal structure of GPX1 protein tetrameric assembly from the Protein Data Bank in Europe database (https://www.ebi.ac.uk/pdbe/entry/pdb/2f8a/analysis, accessed on 1 May 2022).

**Figure 2 cancers-14-02560-f002:**
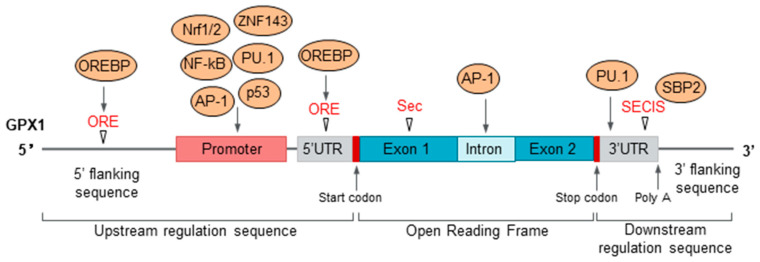
Schematic diagram of GPX1 gene expression and activity regulation.

**Figure 3 cancers-14-02560-f003:**
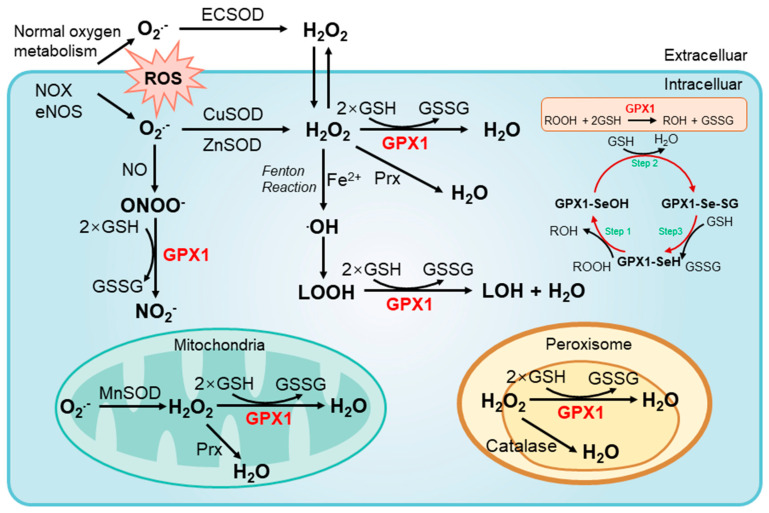
The enzymatic mechanisms of GPX1 to remove superoxides.

**Table 2 cancers-14-02560-t002:** Functional characterization of GPX1 in human cancers.

Tumor Type	Sample Types	Detection Methods	Expression (Tumor vs. Normal)	Roles in Cancer	Functions In Vitro	Functions In Vivo	Ref.
Pancreatic cancer	Patient tissues embedded in paraffin	Immunohistochemistry	Down	Tumor suppressor	—	—	[69]
	Cells (BxPC-3, Capan-1, MIAPaCa-2, AsPC-1)	Western blot, Enzyme activity assay	Down (protein, immunoreactivity)	Tumor suppressor	—	—	[70]
	Cells (MIAPaCa-2, AsPC-1), Mice	—	—	Tumor suppressor	GPX1 overexpression inhibits cell growth, plating efficiency, soft agar plating efficiency	GPX1 overexpression slows tumor growth and increases animal survival	[71]
	Patient frozen tissues Patient tissues embedded in paraffin Cells (PANC-1, MiaPaCa-2, BxPC-3, CFPAC-1, SW1990)	RT-qPCR, Western blot, Immunohistochemistry	Down	Tumor suppressor	GPX1 silencing increases colony formation capacity, EMT phenotype, gemcitabine resistance	Decreased GPX1 expression predicts poor OS, induces EMT phenotype and gemcitabine resistance	[72]
	Cells (MiaPaCa-2, SW1990, PANC-1)	—	—	Tumor suppressor	GPX1 overexpression sensitizes cells to starvation-induced cell death	—	[73]
Oral squamous cell carcinoma	Patient tissues embedded in paraffin	Immunohistochemistry	—	Tumor suppressor	—	High expression of GPX1 predicts better disease-specific survival	[74]
	Patient tissues embedded in paraffin	Immunohistochemistry	Up (Protein)	Tumor promoter	—	GPX1 overexpression predicts nodal metastasis, advanced stage, high grade, invasion and poor survival	[75]
Gastric cancer	Cells (SNU1, 5, 16, 216, 484, 601, 620, 638, 668, 719) Patient tissues embedded in paraffin	RT-qPCR, Immunohistochemistry, Bisulfite sequencing	Down (mRNA, protein) Up (promoter methylation)	Tumor suppressor	—	Decreased GPX1 expression predicts aggressiveness, lymphatic invasion, and poor survival	[76]
Thyroid cancer	Patient fresh tissues	—	Down	Tumor suppressor	—	Decreased GPX1 expression is related to the imbalance of oxidant/antioxidant system	[77]
Kidney cancer	Patient frozen tissues (KIRP)	RNA-sequencing	Up (mRNA)	Tumor suppressor (KIRP)	—	High GPX1 expression predicts better OS	[68]
	Patient frozen tissues (RCC)	RNA-sequencing	Up (mRNA)	Tumor suppressor (KIRP), Tumor promoter (KICH, KIRC)	—	High GPX1 expression predicts poor prognosis in KICH and KIRC, but better OS in KIRP	[78]
	Cells (A-498, ACHN, 786-O, CAKI-1), Patient frozen tissues (RCC)	Western blot, Immunohistochemistry	Up (Protein)	Tumor promoter	GPX1 knockdown inhibits proliferation and clonogenic capacity	GPX1 overexpression predicts poor overall survival, distant metastasis, lymphatic metastasis, and tumor stage	[79]
Breast cancer	Cells (T47D)	—	—	Tumor promoter	GPX1 overexpression inhibits doxorubicin-induced apoptosis	—	[80]
	Cells (MDA-MB-231), Rat mammary tumors	—	—	Tumor promoter	—	Decreased GPX1 activity sensitizes breast cancer cells to anthracyclines	[81]
	Cells (MDA-MB-231, MDA-MB-468, Hs578T, BT-549) Mice	RT-qPCR, Western blot, enzyme activity analysis	Up (mRNA, protein, enzyme activity, in TNBC cells)	Tumor promoter	GPX1 expression promotes migration and invasion	GXP1 silencing reduces lung metastasis of TNBC cells	[82]
	Cells (MDA-MB-231)	—	—	Tumor promoter	GPx1 silencing increases TNF-α-induced apoptosis	GXP1 silencing reduces tumorigenic growth	[83]
	Cells (MDA-MB-231)	—	—	Tumor promoter	GPX1 activated by glutamate dehydrogenase 1 to promote cancer cell proliferation	GPX1 activated by glutamate dehydrogenase 1 to promote cancer cell growth	[84]
Glioma	Glioma stem cells (U87, SU-2)	RT-qPCR, Western blot, Enzyme activity assay	Up (mRNA, protein, enzyme activity)	Tumor promoter	Increased GPX1 expression decreases ROS level, and increases radioresistance and stemness in GSCs	—	[85]
	Patient frozen tissues	RNA-sequencing	Up (mRNA)	Tumor promoter	—	High GPX1 expression predicts poor OS and DFS in LGG	[68]
Hepatocellular carcinoma	Patient fresh tissues	Enzyme activity assay	Up (enzyme activity)	Tumor promoter	—	Increased GPX1 activity correlates with vascular invasion	[86]
Colorectal cancer	Cells (DLD-1)	—	—	Tumor promoter	Increased GPX1 activity protects cancer cells from H_2_O_2_-induced cell death	—	[87]
Esophageal cancer	Cells (EC9706, EC109, K150, K180)	—	—	Tumor promoter	GPX1 overexpression promotes invasion, migration, proliferation, and cisplatin resistance	—	[88]
Salivary adenoid cystic carcinoma	Cells (ACC-M, SACC-83, ACC-2)	—	—	Tumor promoter	GPX1 overexpression promotes proliferation, invasion, migration and cisplatin resistance	—	[89]
Lung cancer	Cells (H1299)	—	—	Tumor promoter	GPX1 activated by glutamate dehydrogenase 1 to promote cancer cell proliferation	GPX1 activated by glutamate dehydrogenase 1 to promote cancer cell growth	[84]
	Cells (A549, H1975, H460, H1650, GLC-82, H1993, H2170, Spc-a1, H1299)	RT-qPCR, Western blot	Up (in cisplatin-resistant cell lines)	Tumor promoter	GPX1 overexpression inhibits ROS accumulation and leads to cisplatin resistance	—	[90]
Laryngeal squamous cell carcinoma	Patient frozen tissues Patient tissues embedded in paraffin	RT-qPCR, Immunohistochemistry	Up	Tumor promoter	—	GPX1 overexpression predicts nodal lymph node metastasis, TNM stage and poor survival	[91]
Acute myeloid leukemia	Patient frozen tissues	RNA-sequencing	Up (mRNA)	Tumor promoter	—	High GPX1 expression predicts poor OS	[68]
	Cells (MV4-11), Patient frozen tissues	RT-qPCR, Western blot, RNA-sequencing	Up (mRNA)	Tumor promoter	GPX1 knockdown inhibits cell viability	High GPX1 expression is associated with poor survival and immunosuppression	[92]
Lymphoma	Cells (DOGUM, GUMBUS)	Western blot	Up (protein, in chemoresistant cells)	Tumor promoter	GPX1 expression increases chemoresistance	—	[93]
Malignant pleural mesothelioma	Patient frozen tissues	RT-qPCR	Up (mRNA)	Tumor promoter	—	High GPX1 expression predicts poor OS	[94]
Prostate cancer	Patient tissues embedded in paraffin	Immunohistochemistry	—	—	—	GPX1 expression has no association with cancer recurrence	[95]

RT-qPCR, quantitative real-time polymerase chain reaction; OS, overall survival; DFS, disease-free survival; EMT, epithelial–mesenchymal transition; ROS, reactive oxygen species, TNM, Tumour-Node-Metastasis; RCC, renal cell carcinoma; KIRP, renal papillary cell carcinoma; KICH, chromophobe carcinoma; KIRC, renal clear cell carcinoma; LGG, brain lower grade glioma; TNBC, triple-negative breast cancer; GSCs, glioma stem cells.

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
