# Peer review of "Glutathione Peroxidase GPX1 and Its Dichotomous Roles in Cancer"

_cancers, 2022, doi:10.3390/cancers14102560_

Round 1

Reviewer 1 Report

Summary

In this article, Zhao and colleagues review data of the association between GPX1 gene polymorphism and susceptibility to cancer, and GPx1 cancer gene expression and cancer prognosis, after a brief summary of GPx1 enzymology and expression. They then address the potential of drugs that inhibit GPx1 enzymatic activity in the treatment of cancer.

General comments

This review article is relatively correctly written, although this aspect could be improved. The question under scrutiny is of relative relevance to cancer, considered not anymore at the front of research. The review is based on a relatively abundant literature, 98 papers, that are mostly adequately chosen. However, when tackling the relevance of a gene to pathogenesis, one should also consider data from gene knockout and overexpression models, which is missing here. Also, when considering the GPx enzyme family, one should pay attention to the difference in enzyme attributes between isozymes. For instance, GPx1 enzymatic activity is specific to H2O2, whereas GPx4 to both H2O2 and lipid hydroperoxides. Also the importance of GPx4 as a sentinel enzyme against ferroptosis should be considered, at least briefly. The review however suffers from the fact that there is no real effort of data synthesis, papers are cited one after the other, often contradicting each other. Synthesis only comes at the end in a small summary paragraph. A non-exhaustive list of questions relating to the sense of confounding sentences or of scientific inexactitude is presented below.

Specific comments

  1. Abstract, lane 29: explain what means “due to the influences from confounders”
  2. Page 1: lane 46: “aerobic biogenesis and oxidative enzymes”, rather say aerobic metabolism; oxidative enzymes is unclear
  3. Page 3. The following citation is unclear, as probably too elliptic:”p53 can simultaneously activate the expressions of H2O2-producing enzyme manganese superoxide dismutase (MnSOD) and antioxidant enzyme GPX1 to induce oxidative stress and apoptosis”: are authors understanding that fighting oxidative stress promote apoptosis? Further, this idea contradicts the next citation “High glucose can induce TBP-associated factor 1 (TAF1)-mediated p53 Thr55 phosphorylation in endothelial cells, which dissociates p53 from GPX1 promoter and leads to GPX1 expression decreases and oxidative stress” that indicate that GPx1 expression protects from apoptosis. Authors should provide more data from the cited papers for readers to understand.

  1. Page 4.

Line 129: there is no translational regulation per se at the level of Sec insertion. The appropriate word could be that GPx1 translation is altered upon experimental selenium depletion and alteration of the SECIS machinery.

Lanes 131-137: redundant with page 2 lanes 83-89

Lanes 139-144: should explain how homocysteine inhibits GPx1 expression: again more details are needed here.

Lane 158: to the knowledge of this reader, endothelial nitric oxide synthase (eNOS) does not or is not a classical producer of the superoxide anion.

Lanes 159-161: the superoxide anion is very instable, and does not accumulate in the compartments cited, as readily converted into H2O2, either spontaneously or enzymatically by the SODs

  1. Page 7

Lane 243: explain what means “lower GPX1 enzyme activity to stimulations”

Lane 262: explain “and found that alleles expressing Pro198Leu and Ala7 polymorphisms 262 were more localized in the cytoplasm than other alleles”

  1. Page 11

Lane 399: “Functional enrichment analysis found that GPX1 is involved in several important biological processes, including ROS metabolic process, ferroptosis, cell growth signal transduction, glutathione 401 metabolism, and p53-regulated metabolic pathways. What this sentence has to do in this paragraph?

The following sentence is unclear: “Moreover, GPX1 may involve in AML 409 immunosuppression for its positive correlations with immunosuppressive cells (e.g. 410 myeloid-derived suppressor cells), and immune inhibitory checkpoint-mediated 411 pathways…”

Reviewer 2 Report

This timely and interesting review by Zhao, et al. reviewed the function and regulation mechanisms of Glutathione peroxidase GPX1. They comprehensively summarized previous findings that investigated the association of GPX1 and human cancers and discussed its dichotomous roles in cancer. The authors also discussed the potential therapeutic strategies that target different regulation machinery of GPX1. Overall, this review is well written and comprehensive. It is suitable for publications with only one thing to be modified.

Please improve the quality of Figure 1b. The authors should remake the figure and summarize the most important messages they want to deliver to the readers and provide this information in this figure.

Author Response

Thank you for the favorable comment from Reviewer 2!

We provide improved Figure 1 as the reviewer’s suggestion. Please check it in the revised manuscript.

Round 2

Reviewer 1 Report

Summary

This paper is a revision of a review article addressing the role and association of GPx1 with cancer.

This review has the virtue of a catalogue of papers that deals with the subject, but still lack sa synthesis on the material reviewed. Noted below of specific comments concerning the clarity of the data presented and their possible lack of coherence.

Specific comments

Page 6

Lane 220 says the mutation may cause a conformational change of the enzyme. This should be linked to what is said above about the loss of 5% in GPx1 enzymatic activity.

Lane 226 “A comprehensive meta-analysis including 31 226 published literature found that GPX1 Pro198Leo polymorphism may promote cancer 227 susceptibility by disturbing the antioxidant balance.” How a meta-analysis can reach a conclusion on cell physiology.

Page 8,

Lane 236: the title paragraph should add susceptibility

lane 237: In 2004, Cox D et al. observed no associations between the Leu198 allele in Pro198Leu 237 polymorphism or 1040A allele in -1040 G/A polymorphism of GPX1 and breast cancer  risks in Caucasian female population” is difficult to understand

Lane 244” and with lower GPX1 enzyme activity to selenium mediated stimulation” unclear

Lane 245: the word “phenomenon” is not fit. Replace with trend or observation

Lanes 250-256: the paragraph is confusing, clarify

Lanes 261-267: the paragraph is out of place, and is confusing

Page 9

Lanes 288-29-: unclear: in which context the poly alanine track was identified?

Page 10, lane 338: Pancreatic cancers have already a very low prognosis

Lanes 359-374: the all paragraph is very confusing, because it is not clear whether GPx1 is pro or anti-cancerous

Lane 365: “GPX1 depletion prevents cell migration and invasion by interacting with FAK kinase and 365 downregulating FAK/c-Src activation in vitro and also reduces lung metastasis in vivo” the sentence does not make sense: if GPx1 is depleted it cannot anymore react with FAK. This all paragraph is in fact very difficult to follow, as it is not clear whether GPx1 is pro- or anti-carcinogenic.

Lane 375: the all paragraph is also very confusion: authors describe results of experiments to amount to anecdotical knowledge on the biological role of GPx1 in cancer, if any. Authors should try synthesizing and organizing these data in something that would be more coherent.

Page 11

Lane 393: the first sentence is an statement that lack proofs for sustaining it, and is then contradicted by the following sentence that says that data are contradictory.

Lane 399: what is the link between functional enrichment and the title of the paragraph that suggest the question of the prognostic values of the GPx1 status in cancer?

Lane 416: similarly the results of a knockdown experiment cannot be included in a paragraph[h dealing with prognosis.

Page 12

Lane 437: again a blunt statement not supported by any objective findings

Author Response

This manuscript is a resubmission of an earlier submission. The following is a list of the peer review reports and author responses from that submission.